# *Algorithmic Fidelity* of Large Language Models in Generating Synthetic German Public Opinions: A Case Study

**Bolei Ma**[*1,2]   **Berk Yoztyurk**[*1]   **Anna-Carolina Haensch**[1,2,3]   **Xinpeng Wang**[1,2]
**Markus Herklotz**[1]   **Frauke Kreuter**[1,2,3]   **Barbara Plank**[1,2]   **Matthias Assenmacher**[1,2]

[1]LMU Munich, [2]Munich Center for Machine Learning,
[3]University of Maryland, College Park

[*]Equal contribution.

**Correspondence:** bolei.ma@lmu.de

## Abstract

In recent research, large language models (LLMs) have been increasingly used to investigate public opinions. This study investigates the *algorithmic fidelity* of LLMs, i.e., the ability to replicate the socio-cultural context and nuanced opinions of human participants. Using open-ended survey data from the German Longitudinal Election Studies (GLES), we prompt different LLMs to generate synthetic public opinions reflective of German subpopulations by incorporating demographic features into the persona prompts. Our results show that Llama performs better than other LLMs at representing subpopulations, particularly when there is lower opinion diversity within those groups. Our findings further reveal that the LLM performs better for supporters of left-leaning parties like *The Greens* and *The Left* compared to other parties, and matches the least with the right-party *AfD*. Additionally, the inclusion or exclusion of specific variables in the prompts can significantly impact the models' predictions. These findings underscore the importance of aligning LLMs to more effectively model diverse public opinions while minimizing political biases and enhancing robustness in representativeness.[1]

## 1   Introduction

Recent advances in LLMs have generated significant interest in their potential for synthetic data generation across various domains. A key and widely debated question is whether LLMs can produce synthetic data that accurately represent human opinions (Argyle et al., 2023; Santurkar et al., 2023; Veselovsky et al., 2023; von der Heyde et al., 2025; Long et al., 2024, *inter alia*).

In social science research, it is usually surveys that provide insights into the attitudes and opinions

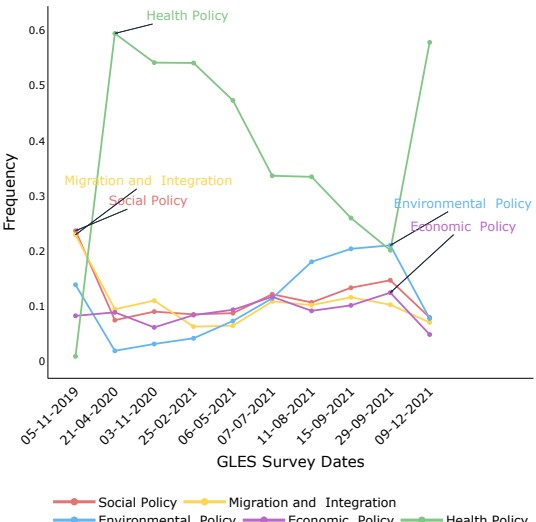

Figure 1: The distribution of the top 5 answer categories between November 2019 and November 2021 in the German GLES survey. There is a significant surge in the Health Policy category from November 2019 to April 2020, with it becoming the dominant focus during this period and afterwards, likely due to the outbreak of COVID-19 in early 2020.

of a population. Recent studies have explored using information from survey responses for LLM prompts, i.e., creating so-called personas and then using the LLM "responses" to additional questions (Argyle et al., 2023; Dominguez-Olmedo et al., 2024; Durmus et al., 2024, *inter alia*). Assessing the fidelity of LLMs in capturing and reproducing human opinions deepens our understanding of model behavior while at the same time helping researchers design more reliable models aligned with human values and improving their usability (Ma et al., 2024). Among these studies, Argyle et al. (2023) introduced the *algorithmic fidelity*, a concept for assessing how effectively LLMs replicate the socio-cultural context and nuanced opinions of diverse human subpopulations. Their work used

---

[1]The code for experiments and evaluation is available at https://github.com/soda-lmu/llm-opinion-german.

*Proceedings of the 63rd Annual Meeting of the Association for Computational Linguistics (Volume 1: Long Papers)*, pages 1785–1809
July 27 - August 1, 2025 ©2025 Association for Computational Linguistics

LLMs to generate synthetic responses ("silicon samples") informed by demographic and ideological profiles from political science datasets. Their findings suggest that LLMs can closely approximate real-world opinion distributions in certain contexts, such as U.S. elections, although challenges persist in fully aligning the generated data with actual demographic groups.

Recent research on LLM responses to opinion polling have predominantly focused on US-based and English-centric survey data. For example, von der Heyde et al. (2025) evaluated the algorithmic fidelity of GPT-3.5 in predicting German voting behavior. Their findings revealed that while GPT-3.5 accurately represented the voting patterns of center and left-leaning political groups, it struggled to capture those of right-leaning parties. However, like many similar studies, their analysis was limited to closed-ended survey questions with single-choice responses. This highlights a key challenge: while LLMs may perform well in English-speaking contexts, less is known about their ability to generate representative opinions in non-English-speaking countries and for open-ended questions. This is particularly true for open-ended setups, where scaling and accurately interpreting responses pose significant challenges (Resnik et al., 2024).

To address these challenges, this study explores the *algorithmic fidelity* of LLMs in generating synthetic public opinions in an open-ended survey question based on German survey data. We use the survey question about the most important problems facing Germany today from the German Longitudinal Election Study (GLES). The survey is a longitudinal panel survey, and the answer distribution can be found in Figure 1. We select variables from the original survey data to represent survey participants with their characteristics as personas. We include three LLMs (Gemma, Llama2, Mixtral) in our study and prompt them to simulate survey participants to answer the open-ended question about the most important political problem in Germany. Finally, we compare the outputs regarding the distributional alignment with original survey answers, predictive performance, and answer diversity. Our most important findings are:

(1) Llama2 is better at modeling group opinions (§4.1).

(2) Llama2's representativeness fluctuates across survey waves, with the model's representativeness of the population decreasing as survey diversity increases; the model represents subpopulation opinions unevenly, with favorable alignment for left-leaning parties (such as the Left, the Greens) over right-parties (such as AfD) (§4.2).

(3) Including more variables in prompts improves performance, with party affiliation being the most influential factor (§4.3).

## 2 Related Work

**LLMs for Survey Response Generation.** Recent studies have increasingly repurposed survey questionnaires, originally designed for public opinion polling, to assess the opinions generated by LLMs (Ma et al., 2024). For instance, Santurkar et al. (2023) identified significant differences between LLM opinion distributions and US-based survey participants. Similarly, Dominguez-Olmedo et al. (2024) highlighted disparities between LLM and human opinions, emphasizing the sensitivity of model outputs to biases in prompting. Tjuatja et al. (2024) found that LLMs are highly sensitive to prompt perturbations and fail to replicate human-like behavior. Collectively, these studies suggest that LLMs align more closely with populations holding left-leaning, Western-oriented values.

**Opinion Generation in the German Contexts.** While most studies on opinions in LLM output are English- and U.S.-centric, some research has explored other contexts, such as the German case. In a recent study, von der Heyde et al. (2025) employed the data of 2017 post-election cross-section of the GLES. Respondents to this study reported their vote choice in the survey. von der Heyde et al. (2025) prompted GPT-3.5 (Brown et al., 2020) with personas to simulate the survey participants. Based on the close-ended choice setup, they found that it does not predict citizens' vote choice accurately, exhibiting a bias towards the Green and Left parties, similar to previous work in English contexts.

**Evaluation of LLM Outputs.** Previous studiesprimarily focused on closed-ended multiple-choice questions, often relying on the model's first token prediction (e.g., Santurkar et al., 2023; Dominguez-Olmedo et al., 2024; Tjuatja et al., 2024) or semi-automated extraction of text answers (von der Heyde et al., 2025). Alternatively, Wang et al. (2024a,b) proposed training a classifier directly on LLM responses manually labeled by

annotators, finding this method more robust. After the output extraction, a few evaluation metrics have been applied to measure the alignment of human and LLM responses (Ma et al., 2024), such as Cohen's Kappa (Argyle et al., 2023; Hwang et al., 2023), 1-Wasserstein distance (Santurkar et al., 2023; Hwang et al., 2023), KL divergence (Dominguez-Olmedo et al., 2024; Sun et al., 2024), Euclidean distance (Wang et al., 2023), Jensen-Shannon distance (Durmus et al., 2024), etc. and correlation and statistical analyses (Sun et al., 2024; Jiang et al., 2024b). For our case study, we adapt these metrics to examine the fidelity of LLM-generated synthetic German public opinions.

## 3 Experimental Setups

### 3.1 Data

**German Longitudinal Election Study (GLES Panel).** We use the GLES Panel dataset from GESIS (2023). The survey consists of 21 waves[2] and contains socio-demographic information, vote intentions, choices, and political attitudes of participants. The target population is German citizens eligible to vote during the respective elections in Germany. Along with the respondents' socio-demographic data for our prompts, we use respondents' answers to the question "In your opinion, what is the most important problem facing Germany today?" for comparing human answers and LLM outputs. The answers of participants were collected as free-form texts.

**Selected Information.** We included six variables from the original survey: age, gender, leaning party, region, education degree, and vocational degree. Details on the sub-groups of the variables are shown in §A.

**Coding Scheme.** For coding the LLM text responses into categories, we follow the coding scheme proposed in an additional sub-study of the GLES (GESIS, 2024). Like Mellon et al. (2024), who collapsed ∼50 classes into a simpler classification, we also set "coarse" classes (n=16). We merged rarely represented classes into an upper class (e.g., "Price Level", "Housing Policy", and "Economic Policy" classes into one "Economic Policy" class). The distribution of GLES survey answers based on the coarse classes is shown in Fig-

ure 1. The full list of fine and coarse classes can be found in §B in the Appendix.

### 3.2 Text Generation

**Models.** We chose three instruction-tuned open-weight LLMs: Llama-2-13b-chat-hf (Touvron et al., 2023), Gemma-7b-it (Team, 2024), and Mixtral-8x7B-Instruct-v0.1 (Jiang et al., 2024a).

**Prompt Design.** After initial trial runs and inspecting the LLM outputs, we used the prompt in Figure 2 in our experiments with LLMs. We chose German as the prompting language, as it is the language in which the GLES survey was conducted, and the generated texts can be compared to the original text. During the experimentation phase, the placeholders are replaced with the respondent's information, i.e., the variables from the survey data.

Identify the most important problem Germany in {month} {year} is facing. Provide the answer in one concise sentence, focusing on a single issue without elaborating or listing additional problems. Do not repeat the information you have been given and give your answer directly and without introductory phrases. Answer in German and only in German, do not use English. Answer from the perspective of a respondent with German citizenship and the characteristics specified below.
*{article} (The) respondent is {age} years old and {gender}. {pronoun} {educational_qualification_clause } and {vocational_qualification_clause} {pronoun2} lives in {region} and mainly supports {party}.*

Figure 2: Translated prompt in English. The original prompt in German is presented in Figure 9 in Appendix.

### 3.3 LLM Output Classification

To evaluate and compare the LLM outputs with human responses, we needed to categorize the responses into specific classes and trained a classifier to code the responses from the LLMs.

**Manual Annotation.** Drawing from Wang et al. (2024a,b), we manually annotated 1,500 LLM outputs, selecting 500 outputs randomly from each LLM. We then trained a classifier based on the manually developed annotation scheme for the LLM outputs. Details on the scheme can be found in §C.

**Classifier Training and Inference.** We fine-tuned the German version of the base BERT (Devlin et al., 2019) classifier on the annotated LLM outputs. The fine-tuned classifier achieves a weighted F1 score of 0.93 on the test set. The classifier is then used to classify all LLM responses.

---

[2]A survey wave refers to a single round of data collection in a panel survey, gathering information from the same participants at multiple points in time (Andreß et al., 2013).

## 3.4 Evaluation Metrics for Experiments

In the context of generative models, representativeness is the model's ability to recover population-level properties of the original data (Eigenschink et al., 2023), i.e., a dimension of algorithmic fidelity. To compare the representativeness of the LLM answers with the original survey data and to measure the association between the variables, we used the following evaluation metrics.

**Jensen-Shannon (JS) Divergence.** JS divergence is a symmetric and normalized measure of divergence derived from KL divergence (Kullback and Leibler, 1951). It is calculated as:

$$\text{JSD}(P \parallel Q) = \frac{1}{2} D_{\text{KL}}(P \parallel M) + \frac{1}{2} D_{\text{KL}}(Q \parallel M) \quad (1)$$

where $M = \frac{1}{2}(P + Q)$ is the mixture distribution of $P$ and $Q$ (Lin, 2006). The JS divergence is bounded between 0 and 1 (when using $log_2$), making it easier to interpret than KL divergence. We use JS Distance, the square root of JS Divergence, as in Durmus et al. (2024), because its bounded range facilitates comparison across different data waves. The JS distance is applied to measure the representativeness of the coded LLM answers compared to the real survey data.

**Entropy.** Entropy measures the variability or uncertainty in a set of outcomes (Jurafsky and Martin, 2024):

$$\text{H}(X) = - \sum_{x \in \mathcal{X}} p(x) \log p(x) \quad (2)$$

We use entropy to assess the diversity of text categories in synthetic and survey data. Lower entropy indicates less variability, meaning fewer bits are needed to represent the information in the data.

**Conditional Entropy.** Conditional entropy measures the remaining uncertainty in variable $X$ when another variable $Y$ is known. It calculates the entropy of $X$ given the distribution of $Y$:

$$H(X \mid Y) = \sum_{y \in \mathcal{A}_Y} P(y) \left[ \sum_{x \in \mathcal{A}_X} P(x \mid y) \log \frac{1}{P(x \mid y)} \right]$$
$$= \sum_{x \in \mathcal{A}_X} \sum_{y \in \mathcal{A}_Y} P(x, y) \log \frac{1}{P(x \mid y)} \quad (3)$$

We use conditional entropy to evaluate how much uncertainty remains about responses in the survey when the subpopulation is known. This helps assess whether the synthetic data captures patterns in specific groups within the population.

**Information Gain.** Also called mutual information. It measures how much information one random variable provides about another. It is calculated as the difference between the entropy of the variable and its conditional entropy given another variable:

$$I(X;Y) = H(Y) - H(Y \mid X) \quad (4)$$

It indicates how much knowing one variable (e.g., $X$) reduces uncertainty about another variable (e.g., $Y$). A higher information gain indicates that knowing one variable reduces uncertainty about another variable. In our experiments, we calculate the population entropy $H(Y)$ and conditional subpopulation entropy $H(Y \mid X)$, where $X$ represents demographic features. We will compute $H(Y)$ and $H(Y \mid X)$ for subpopulations and compare the information gained in survey and LLM data.

**Cramér's V.** This is a measure of association between nominal variables (Cramér, 1999). It is based on Pearson's $\chi^2$ test. However, Cramér's V discounts the value of the $\chi^2$ statistic for both the sample size (N) and the size of the table of counts (minimum of row count or column count minus 1 ) (Holbrook, 2022). It is computed as:

$$V = \sqrt{\frac{\chi^2}{N \cdot \min(r - 1, c - 1)}} \quad (5)$$

We use Cramérs' V to check "pattern correspondence" in LLM outputs. We map each input variable ($X_i$) to the output variable (Y), and check whether the pairwise correlations in survey data are also present in the LLM-generated data.

## 4 Experiments and Results

Three main experiments were conducted on the GLES data. The first evaluated all three LLMs using a single wave, focusing on dataset statistics and representativeness (§4.1). The second extended this analysis across multiple waves with Llama2 to track performance over time (§4.2). The third involved ablation studies to assess how different variables affect representativeness and response diversity (§4.3).

### 4.1 Experiment 1: Model Pre-Experiment in One Wave

After dropping the observations with missing features, we used the survey data from wave 12 (collected between 05-11-2019 and 19-11-2019, i.e., before COVID-19) for the first experiment. All

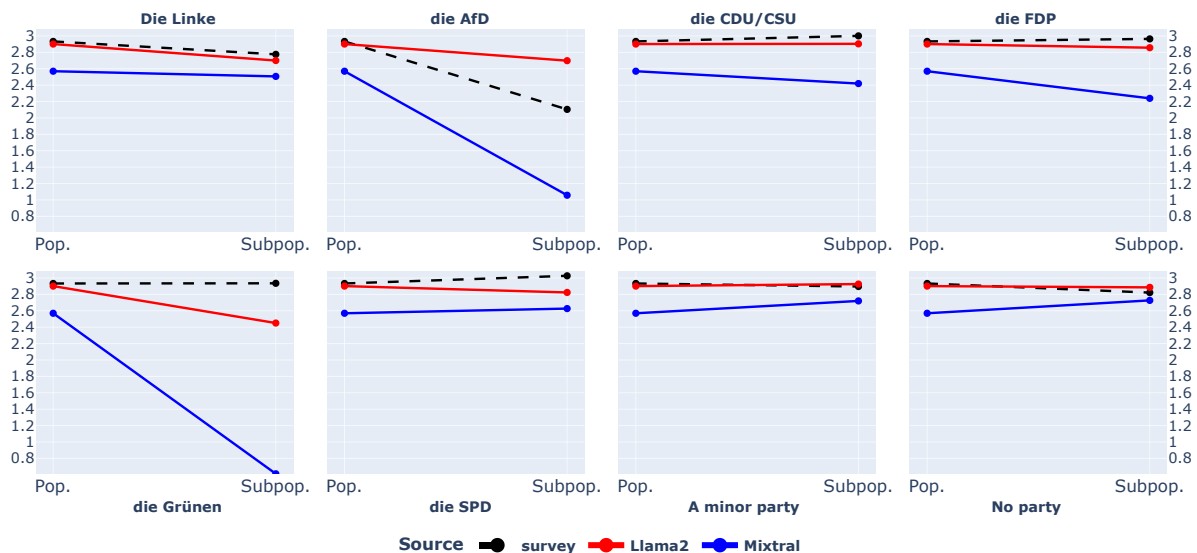

Figure 3: Information Gain for `leaning party` variable ($X_i$). Left: population entropy ($H(Y)$), right: subpopulation entropy ($H(Y \mid X_i)$). A large gap between left and right ($H(Y) - H(Y \mid X_i)$) means big information gain when focusing on the samples of the subpopulation group, indicating responses with this group are less diverse.

three LLMs have been prompted to generate synthetic answers. The generated answers are classified as stated in §3.3. We compared dataset statistics and the textual style of the answers and computed JS Distance and entropy.

In Table 1, we give an **overview of statistics** about labels, textual characteristics, and representativeness (on the population level). 42 % of the Gemma model answers were about COVID-19 (identified using Regex), even though the survey answers were collected before COVID-19, which indicates a very large proportion of hallucinations. Therefore, we did not include the Gemma model in the further subpopulation-level analysis. The detailed JS Distances in each social group category can be found in Table 14 in the Appendix.

| Metric | Gemma | Llama2 | Mixtral | Survey |
|---|---|---|---|---|
| Avg. Labels per Sample | 1.03 | 1.20 | 1.33 | 1.03 |
| Avg. Samples per Label | 593 | 692 | 769 | 597 |
| Avg. Word Count | 36.37 | 25.65 | 43.75 | 2.29 |
| Non-German Answer Rate | 0.02 | 0.06 | 0.03 | - |
| Non-response Rate | 0 | 0 | 0.05 | 0.04 |
| LLM Refusal Rate | 1e-4 | 0 | 1e-4 | - |
| COVID Regex Match Rate | 0.42 | 0.03 | 2e-3 | 0 |
| JS Distance to Survey | 0.62 | **0.28** | 0.29 | - |
| Answer Entropy | 2.26 | **2.90** | 2.56 | **2.93** |

Table 1: Survey and LLM data statistics in experiment 1

**A case study on information gain of the party variables: Llama2 aligns more closely with survey data and maintains subgroup stability.** Figure 3 compares population-level answer entropies (left) with conditional entropies (right) for each `leaning party` value. Information gain, calcu-

lated as the difference between these entropies, reflects how much additional insight is provided by knowing the `leaning party` value. The population-level entropies ($H(Y)$) are close, with the survey (2.93) closely matching Llama2 (2.90), while Mixtral was a bit lower (2.56) (see left of each subplot). After incorporating the `leaning party` information and only looking at the samples containing the specific party affiliations ($H(Y \mid X_i)$), Mixtral shows still lower conditional entropy (see right of each subplot), indicating less variation in responses. Especially for "Die Grünen (The Greens)" and "AfD", there are drastic drops of $H(Y \mid X_i)$. This suggests that Mixtral may risk reflecting dominant group opinions, reducing diversity, and showcasing stereotypical representations of these subgroups. In contrast, Llama2 exhibits less information gain, i.e., it is more aligned with the survey data.

### 4.2 Experiment 2: Wave Experiment with Llama2

We focused on Llama2 in the second in-depth experiment and repeated the generation process for the most 10 recent panel waves in GESIS (2023) (waves 12-21). Over these two years, we observe large shifts in survey label distributions (see Figure 1). This allows us to evaluate the representativeness of the model under varying label distributions and seek answers to the following questions:

- *Do the LLM capabilities at both the population and subpopulation levels vary over time?*

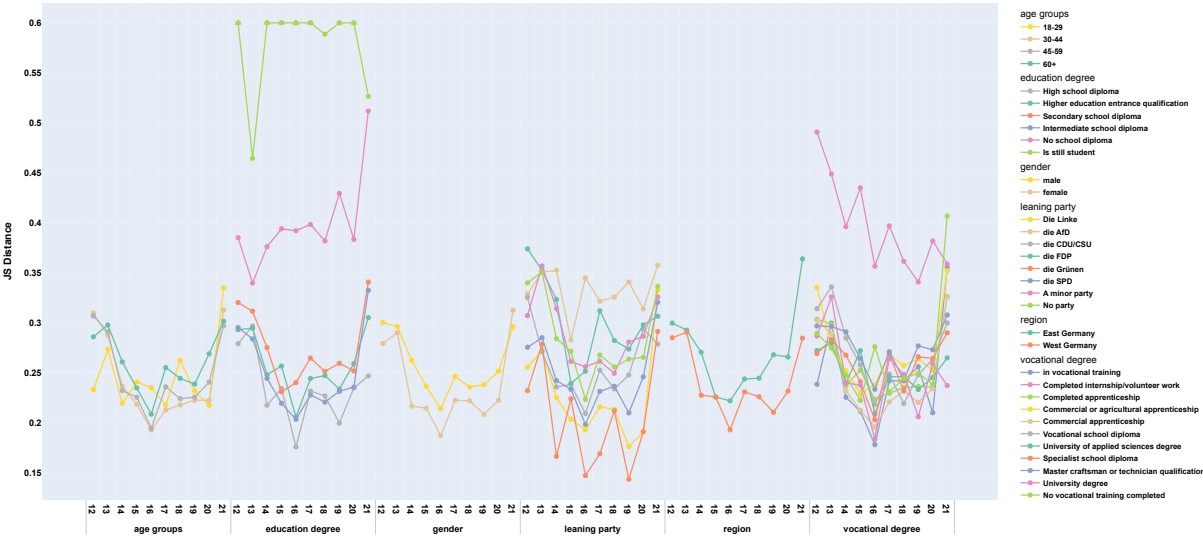

Figure 4: JS Distance of six subpopulation groups in Experiment 2. An in-depth presentation of the JS Distance for each group is shown in Figure 13 in the Appendix.

- *In which subpopulations are opinions represented more accurately?*

**Llama2 captures shifts in survey trends, but higher answer diversity correlates with reduced representativeness.** For panel waves 12-21, we repeated the text generation process and classified the answers. Table 2 shows the population-level entropy values and the JS distances. We calculated Pearson's correlation coefficient between survey entropy and the JS distance and got $r = -0.35$, indicating that the model's representativeness of the population decreases as the diversity in answers increases. For wave 13, with data collection between April 2020 and May 2020, the diversity of answers reached its minimum (with an entropy of 0.58). In Table 13, we see that 92.4 % of answers were about "Health Policy" (and about the COVID-19). This shows that LLMs' responses reflect the change in the survey date.

| | 12 | 13 | 14 | 15 | 16 | 17 | 18 | 19 | 20 | 21 | avg. |
|---|---|---|---|---|---|---|---|---|---|---|---|
| LLM entropy | 2.90 | 0.58 | 1.67 | 1.31 | 2.12 | 2.20 | 2.27 | 2.46 | 2.46 | 2.49 | 2.04 |
| survey entropy | 2.93 | 2.02 | 2.24 | 2.31 | 2.53 | 2.82 | 2.75 | 2.85 | 2.92 | 2.19 | 2.55 |
| JS distance | 0.29 | 0.29 | 0.24 | 0.22 | 0.20 | 0.23 | 0.23 | 0.22 | 0.24 | 0.30 | 0.24 |

Table 2: Population level entropy values and the JS distance in the Wave Experiment from wave 12-21.

**Subpopulation-level findings: JS distances reveal representational variation influenced by group information and complexity.** Figure 4 shows the JS distances at the subpopulation level for each variable. We observed the most variation for education and leaning party variables. Although the difference is smaller than the three variables above, gender and region have a consistent

JS difference regardless of panel waves. And no age variable value consistently has a lower JS score. It shows the model can represent the opinions of different social groups at various levels but offers no clear explanation for the variation in representation. This could come from better recognition of certain groups' views, the training data, or model architecture. Another possibility is that some social groups are more "informative" about this question.

**Llama2 closely reflects sociodemographic patterns, with minor deviations from survey influences.** Cramér's V values in Figure 14 in Appendix show pairwise patterns between prompting variables and text answers. However, in comparison with the survey's values, we see that the model underestimates the influence of age and education degree on the text answers, the model consistently overestimates the effect of region, and the gender and party variables are both overestimated and underestimated. However, except region, the differences are usually less than 0.05, indicating that the Llama2 closely reflects patterns between sociodemographics and the survey.

**A case study on party variables: Llama2 better models groups with left-leaning parties.** To check how much of the JS distance can be associated with the modeling difficulty of the variables, we plot the subpopulation entropy and JS distances for leaning party in Figure 5. Since the population entropy is the same for all subpopulations, we can safely assume that lower conditional entropy means higher information gain for that variable. This allows us to examine how representativeness,

linked to the available information in a variable, impacts alignment success. When mutual information is high, LLMs can better model subpopulation behavior. However, certain groups do not fall into this trend line; "die Linke (the Left)" and "die Grünen (the Greens)" are modeled better, and "AfD" is worse (compared to the information their groups carry). This finding aligns with previous work (e.g., Santurkar et al., 2023; von der Heyde et al., 2025), which shows that LLMs tend to have a more left-leaning feature. We show additional results on other variables and observe similar results showing that LLMs are biased towards left, Western, and educated people in §G (4th paragraph).

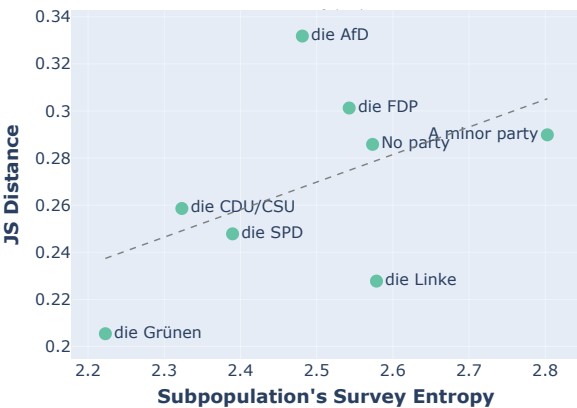

Figure 5: Subpopulation entropy and JS distance for `leaning party` (mean values for waves 12-21).

## 4.3 Experiment 3: Ablation Experiment

To further show how individual demographic information affects the LLM output diversity, we conducted a series of ablation experiments with the following variations:

- Including only one social group variable.
- Excluding one social group variable.
- Using no social group variables.

These were compared against the experiment with all variables included.

In the base prompt, the model was only informed that the response was from a German citizen, with the relevant survey time frame. Detailed prompt variations can be found in Table 8 in Appendix. We used Llama2 and wave 12 data to analyze how adding or removing social group single variables impacts representativeness and answer diversity. The use of wave 12 is because it took place before COVID (see the dates of the waves in §A), and might have more diverse answer categories (compared to the dominance of health policy responses illustrated in Figure 1).

**Variable inclusion and exclusion have an impact on model performance.** Figure 6 shows the JS distances in ablation experiments. Including all variables reduces the JS distance by 0.15 compared to the base prompt. Adding a single variable improves predictions. Removing a variable worsens performance, though it is still better than using only one variable, except for the "all except party" case.

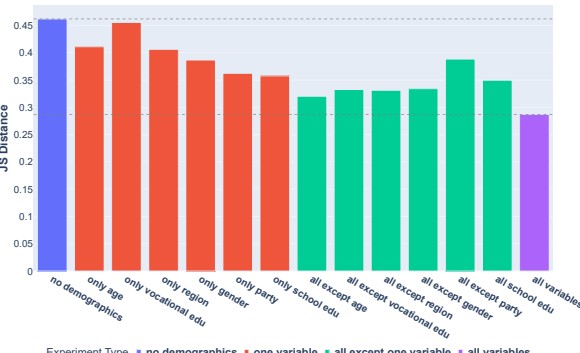

Figure 6: JS distances for the ablation experiment.

**LLM outputs show stronger correlation with variables when prompted with only one variable.** Table 3 compares Cramér's V values between the survey, including only one variable, and all variables included experiments. When only one variable is provided, the generated texts show stronger correlations with the input variable. Although JS distances decrease when more variables are added, this observation suggests that synthetic data patterns are dynamic and can be influenced by the number of prompt variables.

| Prompt Variable | Survey | LLM-one | LLM-all |
|---|---|---|---|
| Age | **0.09** | **0.09** | 0.07 |
| Education Degree | 0.06 | **0.25** | 0.05 |
| Gender | 0.08 | **0.20** | 0.16 |
| Leaning Party | 0.16 | **0.35** | 0.17 |
| Region | 0.06 | **0.42** | 0.15 |
| Vocational Degree | 0.08 | **0.12** | 0.07 |

Table 3: Cramér's V values for the Ablation Experiment

**The inclusion of the `party` variable has the most significant impact on model performance, with its presence leading to substantial improvements in information gain.** As shown in Figure 6, adding either only the party or education variable alone results in the greatest reduction in JS distance compared to the model without demographics; excluding only the party variable leads to a smaller improvement in JS distance, both highlighting the party variable's greater impact. Similar to Figure 3, we plot information gain for `party`, comparing survey entropy to Llama2 with all variables, with only

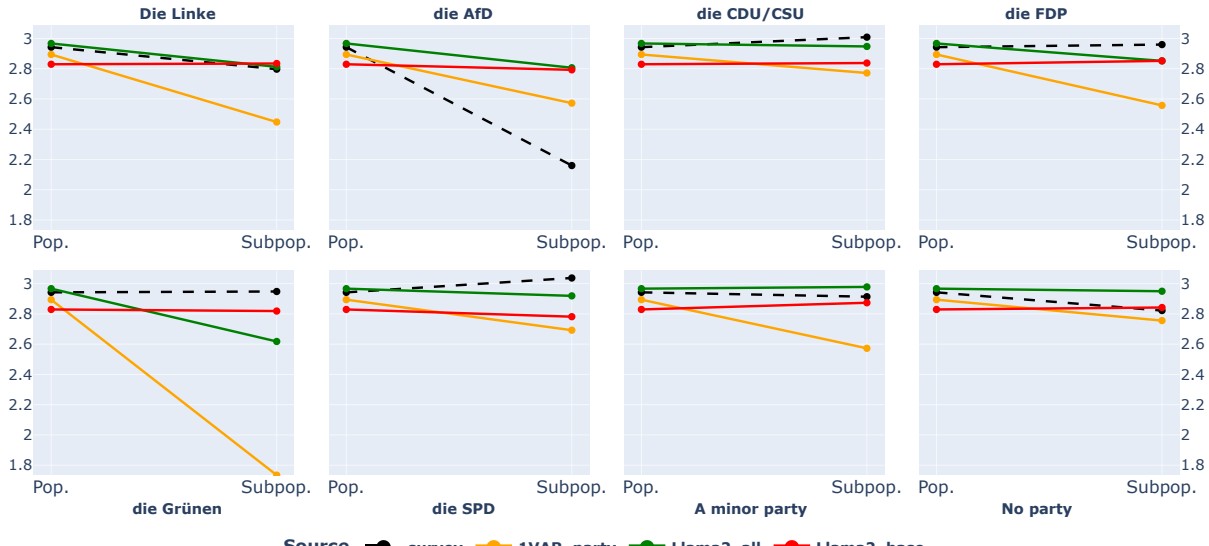

Figure 7: Information Gain for `leaning party` variable, comparing survey entropy to `Llama2-all` (with all variables), `1VAR-party` (with only party variable), and `Llama2-base` (with no variables). Left: population entropy ($H(Y)$), right: subpopulation entropy ($H(Y \mid X_i)$).

one party variable, and with no variables in Figure 7. As expected, `Llama2-base`, which includes no subpopulation variables, produces entropies close to the population entropy, with random variations of $\pm 0.03$. However, in the `1VAR-party` experiment, information gain ranges from 0.2 to 1.3, significantly above random variation. This, along with Cramér's V values, suggests that the model generates typical responses, reducing the variation in subpopulation opinions. Further detailed experimental results are provided in §G of the Appendix.

## 5 Discussion

We next distill key findings from our experiments, compare them to prior research, and offer insights into the role of LLMs in modeling demographic behaviors and their practical insights in survey-based applications based on our German case study.

**Algorithmic Fidelity in Modeling the German population.** von der Heyde et al. (2025) found that GPT-3.5 vote predictions for the 2017 German election are inaccurate and biased towards the Green and Left parties. We also found that the model is better at modeling the opinions of the Green and the Left parties than the right parties. The subpopulation entropy and in-group diversity can partially explain this finding. Other factors could be the models' training data and the RLHF methods used.

**Reduction in in-group diversity.** Bisbee et al. (2024) found that while GPT-3.5 could replicate survey averages, its synthetic answers lacked vari-

ation compared to real survey data. Similarly, von der Heyde et al. (2025) noted GPT-3.5's difficulty in capturing nuanced subpopulation behaviors. In our analysis, we also observe a reduction in in-group diversity under certain conditions, particularly when only one variable is provided to the model or when using the Mixtral model. This suggests that the ability to represent within-group diversity is limited by the model's input structure and specific architecture.

**The role of LLM prompts.** Binz and Schulz (2023) highlighted how cognitive biases, such as the framing effect, influenced GPT-3's outputs. In our third study, we also noticed that providing only one variable in a prompt caused Llama2 to focus disproportionately on that variable, possibly interpreting it as more critical than when multiple variables were included. Interestingly, this effect varied by model: Mixtral over-relied on variables even with full prompts, while Llama2 showed stronger biases with fewer variables. This suggests that model-specific differences influence how demographic factors are integrated and highlights the need for careful prompt design. Argyle et al. (2023) motivated the silicon sampling approach on this conditional probability formula:

$$P(V, B_{\text{LLM}}) = P(V|B_{\text{LLM}})P(B_{\text{LLM}}) \qquad (6)$$

here B is demographic backstories, and V is voting patterns. If the model learned the $P(V|B)$, one could correct for the $P(B)$ and obtain:

$$P(V, B_{\text{survey}}) = P(V|B_{\text{survey}})P(B_{\text{survey}}) \qquad (7)$$

However, in the ablation experiment, we observed that prompting with the social groups is not straightforward, and it does not align LLMs' inner parameters to "solely" consider $P(B_{\text{social\_group}})$. LLMs might not always be conditioned to sample from the joint distribution of backstories. We propose that demographic variables' order, number, and predictive power have a complex interplay and that this is a further research direction (see, e.g., Shu et al., 2024). Also, insights of vignette experiments from survey methodology (Steiner et al., 2017) could be useful in prompt design.

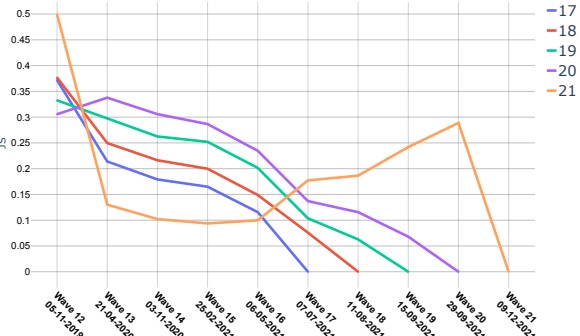

Figure 8: JS distances of answers for the last five waves (17-21) from GLES, comparing each survey's answers to those of the preceding surveys. In most cases, differences between survey responses increase over time. The peak of wave 21 compared to wave 20 corresponds to the drop of the health policy category on 29.09.2021 in Figure 1, possibly due to some effect of COVID related topics in that time.

**Practical evaluation of LLMs.** Figure 8 illustrates the JS distances of responses in the last five survey waves compared to earlier waves. As shown, responses from older surveys tend to differ more significantly from those in recent surveys.

Consider a hypothetical scenario where LLMs are continuously updated with up-to-date training data while surveys are conducted less frequently due to cost constraints. In such a scenario, Figure 8 suggests that LLMs may also help researchers estimate the answers due to the timeliness of their training data. However, even if the model is assumed to be representative enough, the variety in subgroup answers should also be considered for practical uses. In contrast, recent work by Park et al. (2024) demonstrates how LLMs, when applied to large-scale human participant interviews, can simulate subpopulations' attitudes and behaviors, with surveys as a valuable evaluation tool. This highlights the potential of LLMs not only to provide estimates but also to more accurately reflect diverse human perspectives, emphasizing the importance of incorporating subgroup variation.

# 6 Conclusion and Recommendations

This paper evaluates the *algorithmic fidelity* of LLMs to represent the opinions of German subpopulations. While von der Heyde et al. (2025) found that GPT-3.5 struggles with the nuances of German subpopulations and the country's multi-party system in closed-ended voting questions, we explore instead free-form open-ended text responses, focusing on how these responses align with survey data. By using free-form text responses rather than multiple-choice questions, we can identify detailed issues in contextual information and the variety of different subpopulations, underscoring the value of this evaluation approach. Our findings show that LLMs, particularly the Llama2 model, are capable of associating text responses with social-demographic variables, indicating a degree of representativeness. However, the number of variables included in the prompt plays a crucial role in model performance. Despite this, the models still tend to generate stereotypical representations, with a noticeable favor towards left-leaning parties, consistent with previous findings on the limited diversity of opinions reflected in LLMs.

Based on these findings, we recommend that both LLM and social science researchers consider the following steps for future evaluation of LLM-generated responses in survey-based research:

**Improved representation of opinion diversity**: LLMs should be further developed to reflect the full spectrum of opinions within subpopulations while harmful contents are cautiously manipulated. This includes addressing biases and avoiding the over-simplification of diverse views into stereotypical categories with certain safety mechanisms.

**Cross-national comparison**: Due to the current discussion of English-centric biases of LLMs, a more inclusive evaluation with opinion diversity from non-English data or cross-national sources such as GlobalOpinionQA (Durmus et al., 2024) should be conducted and improved.

**Timeliness and survey simulation**: LLMs can be valuable in situations where real survey data is limited or outdated (Namikoshi et al., 2024; Ma et al., 2024). With continuously updated training data, LLMs could be further evaluated in the case of estimating shifts in public opinion.

## Limitations

As in our setup, we need to manually annotate the LLM responses into 16 classes. This limitation constrains our ability to conduct a more comprehensive exploration within this setup, as manual annotation is both costly and inefficient. Future research could address this limitation by exploring the potential of combining human and LLM annotation, as recently proposed by Choksi et al. (2024).

Another limitation concerns the inherent challenge of scaling and coding open-ended survey responses. While we propose reducing the number of classes to 16 in this paper which has also been proposed by recent social science research (Mellon et al., 2024), there might be instances where this simplification may not fully capture the nuanced nature of the responses. To address this, future studies could incorporate qualitative analyses by subject matter experts as part of the final validation process.

Additionally, our study focused on three open-weight LLMs and the German language, which could limit the scope of our findings and might have restricted the generalizability of our findings to other LLMs or languages. Future research should include a broader variety of models and prompting languages to explore their performance and generalizability. Moreover, as our approach primarily relies on zero-shot prompting, a promising next step would be to evaluate how alternative methods, such as fine-tuning (e.g., Cao et al., 2025; Holtdirk et al., 2025), could influence model performance in the German context.

## Ethical Considerations

Throughout the experimentation, we use the publicly available survey dataset from GESIS (2023). The original data contains social-demographics of the human participants of the survey, with the personally identifiable data removed. We solely present the survey results and the responses generated by the LLMs, in an objective manner. We do not intend to express our personal opinions on the questions.

As LLMs are deployed in systems that interact with humans, aligning them with humans' ethical values gains more importance. Although, as of today, LLMs can not reliably support opinion polling surveys, they still carry important information about human opinions, values, and attitudes (Ma et al., 2024). Even if synthetic survey data is not yet used to support surveys, LLMs as LLMs can be important tools for consulting political and social information about countries, populations, sub-populations, and politics in general (Hämäläinen et al., 2023).

When interpreting the opinions reflected in LLM outputs, we advise caution against anthropomorphism. Following recent discussions (Santurkar et al., 2023; Röttger et al., 2024; Durmus et al., 2024; Ma et al., 2024), we use the phrase "opinions reflected in the LLM outputs" rather than suggesting that LLMs "have" opinions, to avoid anthropomorphizing these models.

## Acknowledgments

We thank the members of SODA Lab and MaiNLP labs from LMU Munich, and the members of the Social Data Science Group from University of Mannheim for their constructive feedback. Xinpeng Wang and Barbara Plank are supported by ERC Consolidator Grant DIALECT (101043235). Markus Herklotz and Matthias Assenmacher are supported by the Deutsche Forschungsgemeinschaft (DFG, German Research Foundation) under the National Research Data Infrastructure – NFDI 27/1 - 460037581.

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

# A Data and Prompt Template

Table 4 shows the dates for the waves involved in the original GLES survey (GESIS, 2023). The six main social demographic variables and their subgroups in the original survey are presented in Table 5.

| Wave | Start Date | End Date |
|------|-----------|----------|
| 10 | 06-11-2018 | 21-11-2018 |
| 11 | 28-05-2019 | 12-06-2019 |
| 12 | 05-11-2019 | 19-11-2019 |
| 13 | 21-04-2020 | 05-05-2020 |
| 14 | 03-11-2020 | 17-11-2020 |
| 15 | 25-02-2021 | 12-03-2021 |
| 16 | 06-05-2021 | 19-07-2021 |
| 17 | 07-07-2021 | 20-07-2021 |
| 18 | 11-08-2021 | 24-08-2021 |
| 19 | 15-09-2021 | 25-09-2021 |
| 20 | 29-09-2021 | 12-10-2021 |
| 21 | 09-12-2021 | 21-12-2021 |

Table 4: Data collection dates of GLES waves

In Figure 9, we show the original prompt in German we used for the LLM response generation. We use the template in German and expect the LLMs to respond in German, in order to mirror the real survey case. The placeholders in the prompt are replaced with the respondents' information and then fed to the LLMs that were experimented with. The prompt templates used in the ablation study in §4.3 are presented in Table 8.

| Social Groups | Sub-Groups |
|---------------|------------|
| Age | 18-29 |
| | 30-44 |
| | 45-59 |
| | 60+ |
| Gender | Male |
| | Female |
| Leaning Party | AfD |
| | CDU/CSU |
| | FDP |
| | Grünen |
| | A minor party |
| | Linke |
| | SPD |
| | No party |
| Region | East Germany |
| | West Germany |
| Education Degree | High school diploma |
| | Higher education entrance qualification |
| | Secondary school diploma |
| | Intermediate school diploma |
| | Is still student |
| | No school diploma |
| Vocational Degree | Completed vocational internship/volunteer work |
| | Vocational school diploma |
| | University of applied sciences degree |
| | Specialist school diploma |
| | Completed apprenticeship |
| | Master craftsman or technician qualification |
| | University degree |
| | In vocational training |
| | Commercial or agricultural apprenticeship |
| | Commercial apprenticeship |
| | No vocational training completed |

Table 5: Social-Demographic Groups and Sub-Groups.

Identifizieren Sie das wichtigste Problem, mit dem Deutschland im {month} {year} konfrontiert ist. Geben Sie die Antwort in einem prägnanten Satz an, konzentrieren Sie sich nur auf ein einziges Thema ohne weitere Ausführungen oder Auflistung zusätzlicher Probleme. Wiederholen Sie nicht die Informationen die Ihnen gegeben wurden, und geben Sie Ihre Antwort direkt und ohne einleitende Phrasen. Antworten Sie auf Deutsch und ausschließlich auf Deutsch, verwenden Sie keine Englische Sprache. Antworten Sie aus der Sicht eines Befragten mit deutscher Staatsbürgerschaft und den im nachfolgenden spezifizierten Eigenschaften.
*{artikel} Befragte ist {age} Jahre alt und {gender}. {pronoun} {schulabschluss_clause } und {berufabschluss_clause} {pronoun2} lebt in {region} und unterstützt hauptsächlich {party}.*

Figure 9: Original prompt template in German.

## B Classes of the Coding Scheme

In Table 9, we show the original classes of the coding scheme based on GESIS (2024) as well as the merged and reduced 16 classes.

## C Annotation Scheme

The annotators followed the coding guidelines provided by (GESIS, 2024) for consistency. We used the coarse classes for annotation to achieve better agreement among annotators. One of the leading authors and another project collaborator were asked to conduct the same annotations of the 1,500 samples as volunteers. There exists disagreement on minor cases that were resolved after discussion. Both annotators are consent about the annotated data use. Figure 10 and 11 show the annotation screen and the annotation instruction given to the annotators respectively.

Figure 10: Annotation Screen.

## D Technical Setup

We used Python 3.12.1 and the `transformers` [3] library (version 4.42.4) by HuggingFace (with Pytorch Framework as the backend) to create two custom classes: the `BertClassifier` (for the multilabel classification task) and `TextGenerator` (for generating synthetic answers ).

---

[3] https://huggingface.co/docs/transformers/

Figure 11: The instructions at the annotating tool.

**Text Generation.** To fit models into a single GPU, we have used the 8-bit quantized version of the models. The inference configurations can be found in the study repository. We did the inference with batch_size of 16 to benefit the parallel computing power and reduce runtime. On average, `TextGenerator` generated 1.16 answers per second. We performed 25 generation experiments, using ca. 75 GPU hours for the generation task.

**Text Classification.** As the contexts are all in German, we used the German version of the BERT model[4]. The `BertClassifier` training takes around 20 minutes for the setup 5. We trained the with a batch_size of 32, a learning_rate of 2e-5, and a fixed_precision at 16 bits to fasten the convergence. The early stop condition stopped the training after 3rd epoch when no further loss reduction was observed. Table 6 shows the other relevant model parameters and hyperparameters.

| Parameter | Value |
|---|---|
| epochs | 15 |
| learning_rate | 2e-5 |
| batch_size | 32 |
| weight_decay | 0.01 |
| fp16 | True |
| max_length | 512 |

Table 6: Hyperparameters for the BERT model

## E Qualitative Analysis

Table 10 shows a few sample responses from LLM experiments. Before discussing models' representativeness, we point to three qualitative issues observed in the text generations:

---

[4] https://huggingface.co/google-bert/bert-base-german-cased

**Introductory Phrases.** We noted that all models use some "introductory phrases" even though models were prompted not to use any. The Gemma model starts the sentence 96 % of the time by listing the social group variables given to it. Llama2 model uses the "The most important problem facing Germany" phrase in 96 % of its generations. The Mixtral model uses the "One of the most important issues" phrase in 75 % of the answers. Even if the text lengths had been comparable, these style characteristics would allow humans to discriminate synthetic responses from actual ones. At this point, we did not put further effort into trying different prompts and making the synthetic responses stylistically similar to survey responses.

**Mention of Future Events.** Despite specifying the survey month and year (November 2019), responses occasionally referenced events that occurred after the survey date, such as the COVID-19 pandemic (2020) and the Energy Crisis in Germany (2021). For example, the Gemma model contained COVID-19-related words (COVID, corona, coronavirus, COVID-19, sars-cov, etc.) in 42 % of its responses. This problem was observed relatively less in Llama2 and Mixtral models (3 % and 0.2 % of answers, respectively).

**Mixed-Language Answers.** Although models were instructed to respond in German, a small percentage (up to 3%) of answers had some parts in German, whereas some terms or clauses were in English.

## F   Additional Metrics

In this section, we present metrics in addition to those in §3.4. These include the base metrics for the main experimentation, as well as additional metrics used for the additional results in §G.

**Proportion Agreement.** It is the proportion of two variables exactly matching. Like accuracy, this measure does not consider the probability of matching by chance and should be used as a descriptive quantity (Argyle et al., 2023).

**Cohen's Kappa** ($\kappa$)**.** It is a measure of agreement between two categorical variables and is often used as a measure of inter-rater agreement (Cohen, 1960). Unlike proportional agreement, it corrects for the agreement by chance, and It is defined as $\kappa = \frac{p_o - p_e}{1 - p_e}$ where

- $p_o$ is the observed agreement ratio

- $p_e$ is the expected agreement when annotators assign labels randomly.

**Kullback-Leibler Divergence.** It also known as relative entropy, is a method used in measuring the statistical distance between two probability distributions (Kullback and Leibler, 1951). For distributions P and Q of a discrete random variable $X = [X_1, ..., X_n]$, the Kullback-Leibler (KL) divergence can be defined as:

$$D_{\text{KL}}(P \parallel Q) = \sum_{x \in \mathcal{X}} P(x) \log \left( \frac{P(x)}{Q(x)} \right) \quad (8)$$

KL Divergence is not a distance measure since it does not satisfy the symmetry requirement of a metric. i.e KL(P,Q) != KL(Q,P) , unless P and Q are equal. If not, KL divergence is always greater than 0 and not bounded.

**Absolute Percentage Error (APE).** JS distance enables us to compare model performance at the question level. We employ the APE to evaluate the accuracy of predictions in each category. APE is calculated by determining the absolute differences between predicted and actual frequencies and then normalizing these differences by the reference survey frequencies. For each label, we compute:

$$\text{APE}_L = \left| \frac{y_t - \hat{y}_t}{y_t} \right| \quad (9)$$

where $y_t$ is survey frequency and $\hat{y}_t$ LLM output frequency for the label $l$.

## G   Additional Results

In this section, we show additional results and figures for the main experimentation in §4.

**Label Distribution on LLM Outputs.** Figure 12 shows the label distribution on the three LLM outputs based on the coarse labels.

**Detailed JS Distances of Subpopulation in Experiment 1: Llama2 achieves better performance in most categories.** We show the detailed JS Distances in each social group category in Experiment 1 in Table 14 for all three experimented LLMs. Among the three LLMs, we notice Llama2 has the least JS Distances across the most subcategories compared to the other two LLMs, showing more alignment with the real survey data.

**Additional JS Distances of Subpopulation Variables in Experiment 2: Further indication of WEIRD bias of LLMs.** We show additional results of the survey's subpopulation entropy and the JS Distance between the Llama output and survey results across 10 waves in experiment 2 in Figure 15, i.e., results for four other variables in addition to the variable leaning_party in Figure 5 from §4.2. For the vocational degree variable, groups with a completed vocational internship are the least well-represented on average. For the education degree variable, groups with no degree or only a secondary school diploma are less represented compared to those with higher educational qualifications. Among age groups, older cohorts are less well-represented than younger ones. For the regional variable, which includes only two groups, there is a greater discrepancy in representing East Germany compared to West Germany. Overall, these findings align with prior evidence that LLMs exhibit biases favoring Western, younger, and more educated subpopulations, commonly referred to as WEIRD bias[5], as highlighted in studies such as Santurkar et al. (2023), Cao et al. (2023), Arora et al. (2023), and Agarwal et al. (2024).

**Label-Level Percentage Errors in Experiment 3.** Table 7 shows label-level percentage errors. We compared the best-performing models in the 1-var-* and w/o.-* experiments. In 7 out of 14 labels, using all variables produced the lowest percentage errors. Both JS distances and percentage errors decreased with the inclusion of more variables, reinforcing the representational accuracy of the Llama2 model for the German population.

**Proportional Agreement (PA) and $\kappa$ Scores across Waves: LLMs face in achieving consistent agreement with survey data over time, particularly in representing complex social phenomena.** The scores in Table 11 compare Llama2-generated responses with resampled survey data across different waves. The PA for Llama2 fluctuates significantly, reaching as high as 0.56 in wave 13 but dropping in later waves, especially after wave 17, indicating inconsistencies in the model's ability to align with the original survey data. By contrast, the survey resample maintains relatively

---

The term "WEIRD bias" originates in psychology, where values from Western, Educated, Industrialized, Rich, and Democratic (W.E.I.R.D.) societies are assumed to represent universal "human" values (Sanches de Oliveira and Baggs, 2023).

| Labels | LLM-base | min(1var. *) | min(w/o. *) | LLM-all |
|---|---|---|---|---|
| Political System, Processes | 7.92 | -3.62 | **0.92** | 9.71 |
| Social Policy | -23.76 | -21.94 | -15.16 | **-7.45** |
| Health Policy | 22.02 | 3.32 | 2.84 | **2.08** |
| Family and Gender Equality | -0.27 | -0.25 | -0.14 | **-0.10** |
| Education Policy | -2.20 | -2.15 | -1.88 | **-1.83** |
| Environmental Policy | 4.43 | 0.42 | **0.15** | 2.52 |
| Economic Policy | 8.15 | **7.92** | 11.49 | 9.21 |
| Security | **-6.26** | -7.37 | -7.06 | -6.88 |
| Foreign Policy | 6.58 | 1.93 | 1.59 | **0.54** |
| Media and Communication | 0.09 | -0.01 | **0.00** | 0.03 |
| Others | -3.11 | -3.12 | -3.12 | **-3.00** |
| Migration and Integration | -17.16 | -21.40 | -19.80 | **-14.61** |
| East Germany | **0.01** | -0.18 | -0.17 | 0.14 |
| Values, Polit. Culture, Social | 3.56 | **1.31** | 9.90 | 9.65 |

Table 7: Percentage errors in the ablation experiment. LLM-base denotes no demographics; min(1var. *) denotes best-performing experiment with one variable; min(w/o. *) denotes best-performing experiment with all except one variable; LLM-all denotes all variables.

stable PA scores, ranging between 0.34 and 0.55, indicating better agreement with the original survey. The $\kappa$ scores, however, are low for both Llama2 and the survey resample, with Llama2 performing particularly poorly (0.01–0.04). This suggests that while the model can capture some high-level agreement (as seen in PA scores), it struggles to replicate the nuanced variability and structure of human responses across waves, especially as the diversity of responses increases. These findings underscore the challenges LLMs face in representing complex social phenomena in survey data, especially over time.

**Label-level Breakdown with APE: LLM predictions show over- and underrepresentation of certain political topics compared to survey data.** Table 12 and 13 show the comparison of predicted label percentages between survey and LLM answers in experiment 1 and 2 respectively. In Table 12, we observe that "Security", "Migration and Integration" and "Social Policy" topics are less represented than the survey in all LLM-texts and "Values, political culture and general social criticism" represented much more, with a mean APE=218.1. the Mixtral model emphasized "Environment Politics", whereas Llama2 focused on "Political Systems and Processes" more than others. Table 13 shows "Health Policy", "Values, political culture and general social criticism" and "Economic Policy" are consistently more represented as the text

1800

answer categories, whereas "Migration and Integration" and "Security" are less represented. We also calculated the mean APE per label to compare errors on average on which categories the Llama2 represented the political topics more accurately.

**Factual Knowledge of Llama2 without Demographic Prompts.** We also check actual knowledge of Llama2 model without giving any survey contexts in Table 15. It shows Llama2's general accuracy in providing vote percentages for major elections, though minor errors and formatting issues occur, especially in the 2020 U.S. election. For Germany's most important problem in 2019, Llama2 identifies key issues like climate change and economic security but gives rough percentage estimates rather than precise data. These results suggest that while the model captures broad trends, it struggles with exact figures when not given specific demographic or contextual prompts.

| Experiment Name | Prompt Information |
|---|---|
| 1_var_region | Der/Die Befragte lebt in {eastwest}. [/INST] |
| 1_var_party | Der/Die Befragte unterstützt hauptsächlich {party}. [/INST] |
| 1_var_education_degree | Der/Die Befragte {schulabschluss_clause} [/INST] |
| 1_var_age | Der/Die Befragte ist {age} Jahre alt. [/INST] |
| 1_var_gender | {artikel} Befragte ist {gender} [/INST] |
| 1_var_vocational_degree | Der/Die Befragte {berufabschluss_clause} [/INST] |
| without_age | {artikel} Befragte ist {gender}. {pronoun} {schulabschluss_clause} und {berufabschluss_clause} {pronoun2} lebt in {eastwest} und unterstützt hauptsächlich {party}. [/INST] |
| without_region | {artikel} Befragte ist {age} Jahre alt und {gender}. {pronoun} {schulabschluss_clause} und {berufabschluss_clause} {pronoun2} unterstützt hauptsächlich {party}. [/INST] |
| without_vocational_degree | {artikel} Befragte ist {age} Jahre alt und {gender}. {pronoun} {schulabschluss_clause} {pronoun2} lebt in {eastwest} und unterstützt hauptsächlich {party}. [/INST] |
| without_education_degree | {artikel} Befragte ist {age} Jahre alt und {gender}. {pronoun} und {berufabschluss_clause} {pronoun2} lebt in {eastwest} und unterstützt hauptsächlich {party}. [/INST] |
| without_party | {artikel} Befragte ist {age} Jahre alt und {gender}. {pronoun} {schulabschluss_clause} und {berufabschluss_clause} {pronoun2} lebt in {eastwest}. [/INST] |
| without_gender | Der/Die Befragte ist {age} Jahre alt. Er/Sie {schulabschluss_clause} und {berufabschluss_clause} Er/Sie lebt in {eastwest} und unterstützt hauptsächlich {party}. [/INST] |

Table 8: Ablation experiments and the modified prompt contents. `1_var_*` denotes the experimentation of prompting with only one variation*. `without_*` denotes the experimentation of prompting with all variables except *.

| Fine Labels | Coarse Labels |
|---|---|
| Election Campaign and Government Formation, Political Structures and Processes, Democracy, Bureaucracy, Lobbyism, Corruption | Political System and Processes |
| Values, political culture and general social criticism | Values, political culture and general social criticism |
| Social Policy, Social Justice, Poverty, Unemployment and Basic Security, Pensions and Demographic Change | Social Policy |
| Health Policy, Nursing, Corona Pandemic | Health Policy |
| Family Policy, Gender Equality | Family and Gender Equality Policy |
| Education Policy, School Policy | Education Policy |
| Energy Policy, Environmental Policy, Climate Policy, Natural Disasters | Environmental Policy |
| Economic Policy, Price Level, Infrastructure, Digital Infrastructure, Transport Policy, Housing Policy | Economic Policy |
| Terrorism, Internal Security, Crime and Violence, Radicalization and Extremism, Law and Justice, Defense | Security |
| Foreign Policy, Europe and European Union, (Germany's Relations with) Russia, (Germany's Relations with) Turkey, International Conflicts and Peace, War in Ukraine | Foreign Policy |
| Media | Media and Communication |
| Others | Others |
| Migration and Integration | Migration and Integration |
| East Germany | East Germany |
| Not specified | Not specified |
| Don't know | Don't know |

Table 9: Description of the Classes, originally in German, translated to English

| Model | Original Answer | Translation |
|---|---|---|
| Gemma | Als 61-jährige Frau mit deutscher Staatsbürgerschaft, die in Westdeutschland lebt und hauptsächlich keine Partei unterstützt, ist das wichtigste Problem, mit dem Deutschland im November 2019 konfrontiert ist, die weiterhin bestehende Situation der COVID-19-Pandemie. | As a 61-year-old woman with German citizenship, living in West Germany and mainly not supporting any political party, the most important issue facing Germany in November 2019 is the ongoing situation of the COVID-19 pandemic. |
| Llama2 | Das wichtigste Problem, das Deutschland im November 2019 konfrontiert, ist die sich erhöhte Unsicherheit und Angst in der Gesellschaft, insbesondere bei den Menschen, die sich nicht in der Mitte der Gesellschaftskontaktiven Gruppenzugehörigkeit finden. | The most important problem facing Germany in November 2019 is the growing insecurity and fear in society, especially among people who do not belong to the centre of society. |
| Mixtral | Eines der wichtigsten Probleme, mit denen Deutschland im November 2019 konfrontiert ist, ist der Umgang mit den Folgen des Klimawandels und die Reduzierung von Treibhausgasemissionen, was auch entscheidend für die zukünftige Wirtschaft ist. | One of the most important issues facing Germany in November 2019 is dealing with the consequences of climate change and reducing greenhouse gas emissions, which is also crucial for the future economy. |

Table 10: Sample Answers from the Model Experiment

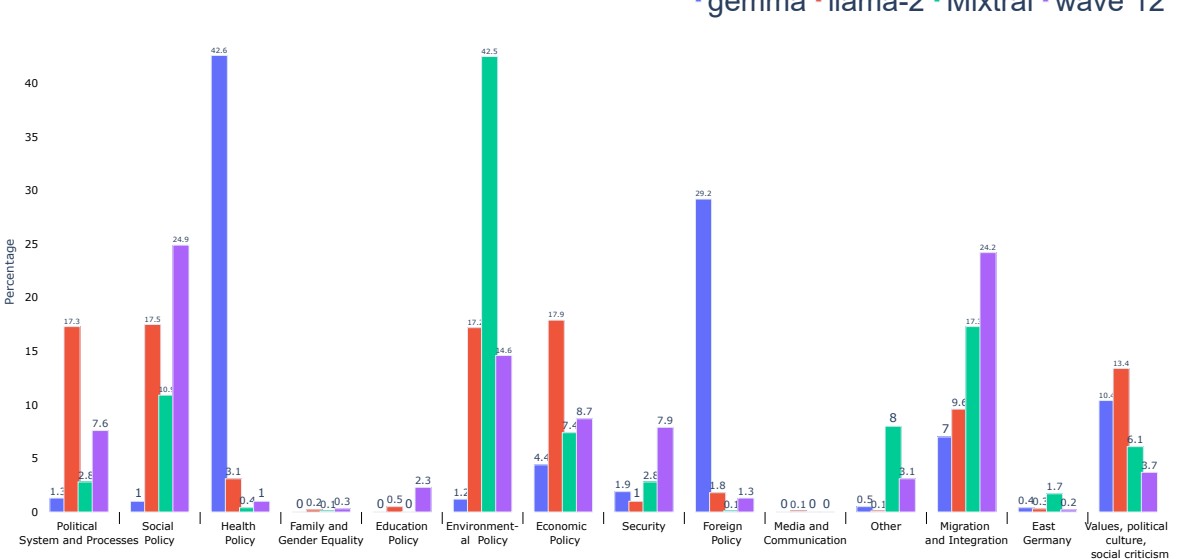

Figure 12: Label distributions of three experimented LLMs

| wave | 12 | 13 | 14 | 15 | 16 | 17 | 18 | 19 | 20 | 21 |
|---|---|---|---|---|---|---|---|---|---|---|
| PA survey resample | 0.38 | 0.55 | 0.52 | 0.46 | 0.42 | 0.37 | 0.38 | 0.36 | 0.34 | 0.52 |
| $\kappa$ survey resample | 0.27 | 0.27 | 0.29 | 0.21 | 0.22 | 0.23 | 0.24 | 0.24 | 0.23 | 0.25 |
| PA Llama2 | 0.14 | 0.56 | 0.41 | 0.46 | 0.31 | 0.24 | 0.23 | 0.21 | 0.19 | 0.25 |
| $\kappa$ Llama2 | 0.02 | 0.01 | 0.03 | 0.03 | 0.03 | 0.02 | 0.02 | 0.04 | 0.03 | 0.02 |

Table 11: Proportional Agreement (PA) and ($\kappa$) Scores. The original survey is the first annotator, and the second annotator is the survey resample (comparison to stratified sampling from the original survey) or Llama2 model

| Category | Source | Gemma | Llama2 | Mixtral | Mean APE |
|---|---|---|---|---|---|
| East Germany | LLM | 0.5 | 0.7 | 1.8 | 368.07 |
| | Survey | 0.2 | 0.2 | 0.2 | |
| Economic Policy | LLM | 4.9 | 20.2 | 14.8 | 78.02 |
| | Survey | 9.0 | 9.0 | 9.0 | |
| Education Policy | LLM | 0.1 | 0.5 | 0.0 | 91.38 |
| | Survey | 2.4 | 2.4 | 2.4 | |
| Environmental Policy | LLM | 1.2 | 14.8 | 35.3 | 78.09 |
| | Survey | 14.6 | 14.6 | 14.6 | |
| Family and Gender Equality Policy | LLM | 0.1 | 0.4 | 0.1 | 56.62 |
| | Survey | 0.3 | 0.3 | 0.3 | |
| Foreign Policy | LLM | 28.8 | 2.0 | 0.3 | 753.35 |
| | Survey | 1.3 | 1.3 | 1.3 | |
| Health Policy | LLM | 41.6 | 3.3 | 0.4 | 1344.72 |
| | Survey | 1.1 | 1.1 | 1.1 | |
| Media and Communication | LLM | 0.1 | 0.1 | 0.0 | 83.89 |
| | Survey | 0.0 | 0.0 | 0.0 | |
| Migration and Integration | LLM | 7.2 | 8.6 | 14.9 | 57.41 |
| | Survey | 24.1 | 24.1 | 24.1 | |
| Others | LLM | 0.0 | 0.2 | 6.1 | 97.60 |
| | Survey | 3.0 | 3.0 | 3.0 | |
| Political System and Processes | LLM | 1.4 | 15.7 | 2.6 | 84.93 |
| | Survey | 7.5 | 7.5 | 7.5 | |
| Security | LLM | 2.0 | 1.8 | 3.3 | 70.10 |
| | Survey | 7.9 | 7.9 | 7.9 | |
| Social Policy | LLM | 1.2 | 16.0 | 12.2 | 60.48 |
| | Survey | 24.8 | 24.8 | 24.8 | |
| Values, Political Culture, and Social Criticism | LLM | 10.8 | 15.6 | 8.3 | 207.01 |
| | Survey | 3.8 | 3.8 | 3.8 | |
| APE | | 150.0 | 70.0 | 73.0 | |

Table 12: Comparison of Predicted Label Percentages in Experiment 1. Colors indicate differences between LLM and survey: green (LLM > Survey + 1%), red (LLM < Survey − 1%), black (|LLM − Survey| < 1%).

| Category | src | 12 | 13 | 14 | 15 | 16 | 17 | 18 | 19 | 20 | 21 | mean APE |
|---|---|---|---|---|---|---|---|---|---|---|---|---|
| Foreign Policy | llm | 1.8 | 0.1 | 0.5 | 0.3 | 1.0 | 0.8 | 1.0 | 1.2 | 1.4 | 1.2 | 112.94 |
|  | survey | 1.2 | 0.3 | 0.3 | 0.3 | 0.3 | 0.4 | 0.6 | 0.7 | 0.4 | 0.4 | |
| Education Policy | llm | 0.5 | 0.2 | 0.2 | 0.4 | 0.6 | 0.4 | 0.4 | 0.4 | 0.4 | 0.6 | 75.13 |
|  | survey | 2.2 | 1.0 | 1.1 | 1.4 | 2.0 | 2.4 | 1.7 | 2.0 | 2.2 | 1.1 | |
| Family and Gender Equality Policy | llm | 0.2 | 0.0 | 0.1 | 0.0 | 0.1 | 0.0 | 0.0 | 0.1 | 0.1 | 0.1 | 61.72 |
|  | survey | 0.3 | 0.1 | 0.1 | 0.3 | 0.2 | 0.3 | 0.2 | 0.2 | 0.2 | 0.2 | |
| Health Policy | llm | 3.1 | 92.4 | 68.9 | 78.5 | 55.6 | 50.6 | 48.3 | 33.9 | 34.0 | 32.0 | 60.4 |
|  | survey | 1.0 | 59.5 | 54.2 | 54.2 | 47.4 | 33.7 | 33.5 | 26.0 | 20.2 | 57.9 | |
| LLM Refusal | llm | 0.0 | 0.0 | 0.0 | 0.0 | 0.0 | 0.0 | 0.0 | 0.0 | 0.0 | 0.0 | nan |
|  | survey | 0.0 | 0.0 | 0.0 | 0.0 | 0.0 | 0.0 | 0.0 | 0.0 | 0.0 | 0.0 | |
| Media and Communication | llm | 0.1 | 0.0 | 0.0 | 0.0 | 0.0 | 0.0 | 0.0 | 0.0 | 0.0 | 0.0 | 98.33 |
|  | survey | 0.0 | 0.0 | 0.1 | 0.1 | 0.1 | 0.0 | 0.0 | 0.0 | 0.0 | 0.1 | |
| Migration and Integration | llm | 9.6 | 0.5 | 1.6 | 1.4 | 2.1 | 2.3 | 2.4 | 2.2 | 2.1 | 1.8 | 77.65 |
|  | survey | 23.0 | 9.5 | 11.1 | 6.4 | 6.5 | 10.9 | 10.3 | 11.7 | 10.3 | 7.1 | |
| East Germany | llm | 0.3 | 0.1 | 0.4 | 0.1 | 0.2 | 0.4 | 0.3 | 0.3 | 0.3 | 0.4 | 949.73 |
|  | survey | 0.2 | 0.0 | 0.0 | 0.0 | 0.0 | 0.0 | 0.0 | 0.0 | 0.1 | 0.1 | |
| Political System and Processes | llm | 17.3 | 0.3 | 1.5 | 1.1 | 1.5 | 1.1 | 1.4 | 1.5 | 1.5 | 1.9 | 75.8 |
|  | survey | 7.2 | 2.7 | 3.0 | 5.2 | 5.4 | 4.6 | 4.6 | 4.5 | 6.7 | 3.2 | |
| Security | llm | 1.0 | 0.2 | 0.2 | 0.3 | 0.6 | 0.7 | 0.5 | 0.4 | 0.4 | 0.4 | 83.5 |
|  | survey | 7.5 | 1.8 | 4.0 | 2.4 | 2.7 | 3.0 | 2.2 | 2.2 | 2.1 | 2.3 | |
| Others | llm | 0.1 | 0.0 | 0.0 | 0.0 | 0.0 | 0.0 | 0.0 | 0.0 | 0.0 | 0.0 | 99.37 |
|  | survey | 3.0 | 2.1 | 2.1 | 3.2 | 3.1 | 2.6 | 2.5 | 2.2 | 2.3 | 1.8 | |
| Social Policy | llm | 17.5 | 1.2 | 9.5 | 5.2 | 14.0 | 16.3 | 15.1 | 17.0 | 17.4 | 26.3 | 56.05 |
|  | survey | 23.7 | 7.5 | 9.1 | 8.6 | 8.8 | 12.2 | 10.7 | 13.4 | 14.8 | 8.0 | |
| Environmental Policy | llm | 17.2 | 0.5 | 2.3 | 3.2 | 8.1 | 10.5 | 12.7 | 23.4 | 22.6 | 14.8 | 30.98 |
|  | survey | 13.9 | 2.0 | 3.2 | 4.2 | 7.4 | 11.5 | 18.1 | 20.5 | 21.1 | 7.8 | |
| Values, Political Culture and General Social Criticism | llm | 13.4 | 1.3 | 5.8 | 2.6 | 4.5 | 3.7 | 4.7 | 4.7 | 4.5 | 5.8 | 191.96 |
|  | survey | 3.6 | 0.8 | 1.4 | 1.4 | 2.0 | 1.5 | 1.4 | 1.4 | 1.4 | 2.1 | |
| Economic Policy | llm | 17.9 | 3.1 | 9.0 | 6.9 | 11.8 | 13.2 | 13.2 | 14.9 | 15.3 | 14.8 | 59.31 |
|  | survey | 8.3 | 8.9 | 6.2 | 8.4 | 9.4 | 11.8 | 9.2 | 10.2 | 12.5 | 4.9 | |
| Not Specified | llm | 0.0 | 0.0 | 0.0 | 0.0 | 0.0 | 0.0 | 0.0 | 0.0 | 0.0 | 0.0 | 100.0 |
|  | survey | 3.7 | 3.0 | 3.5 | 3.1 | 3.8 | 4.0 | 4.0 | 4.1 | 4.9 | 2.4 | |
| Do Not Know | llm | 0.0 | 0.0 | 0.0 | 0.0 | 0.0 | 0.0 | 0.0 | 0.0 | 0.0 | 0.0 | 100.0 |
|  | survey | 1.1 | 0.7 | 0.8 | 0.8 | 0.9 | 1.1 | 0.9 | 1.0 | 0.8 | 0.7 | |
| APE |  | 71.0 | 66.0 | 46.0 | 50.0 | 38.0 | 50.0 | 53.0 | 47.0 | 51.0 | 80.0 | |

Table 13: Comparison of Predicted Label Percentages between survey and LLM-answers in Experiment 2. Colors indicate differences between LLM and survey: green (LLM > Survey + 1%), red (LLM < Survey − 1%), black (|LLM − Survey| < 1%).

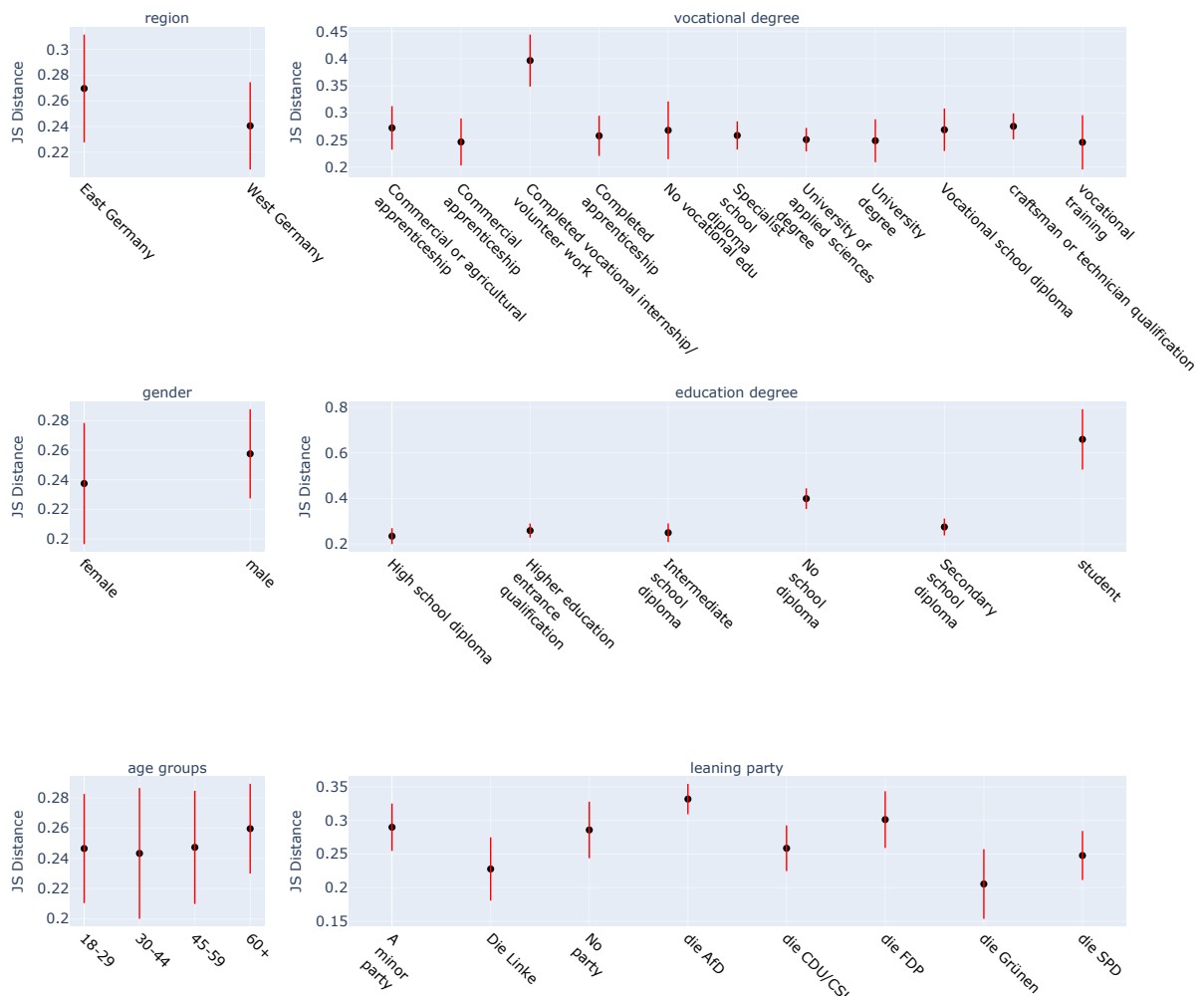

Figure 13: The mean and ±1 standard deviation of JS Distances for social groups in Experiment 2.

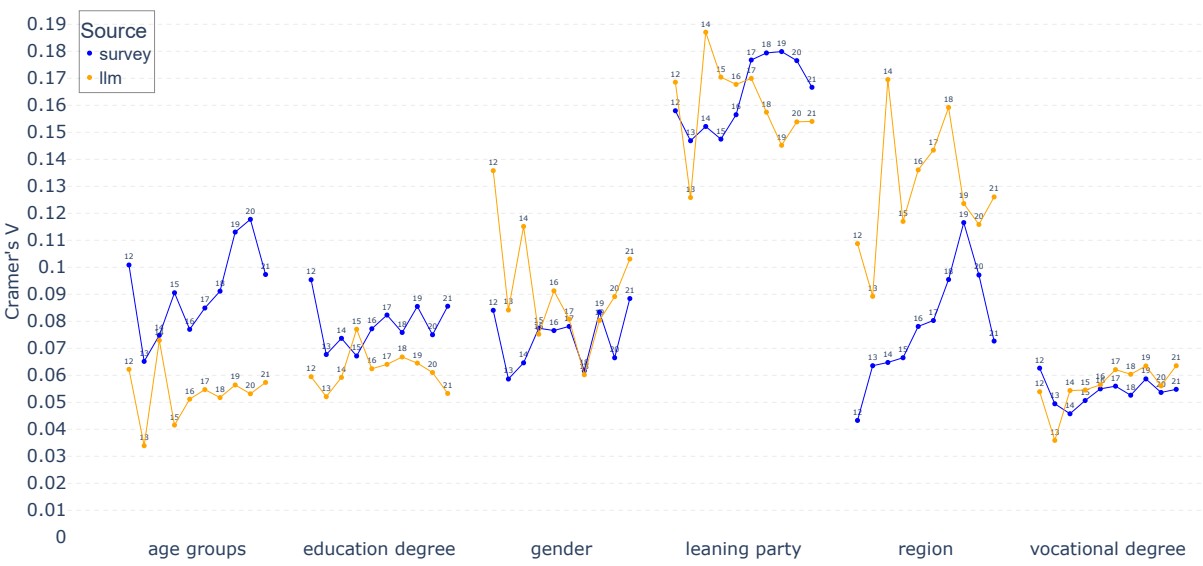

Figure 14: Cramér's Values for pairwise patterns between the six prompting variables and text answers from the survey and LLMs in Experiment 2

| Social Group Category | Social Group | Gemma | Llama2 | Mixtral |
|---|---|---|---|---|
| Population | Population | 0.617 | 0.287 | 0.295 |
| Age Groups | 18-29 | 0.638 | 0.233 | 0.246 |
| | 30-44 | 0.613 | 0.310 | 0.310 |
| | 45-59 | 0.627 | 0.307 | 0.309 |
| | 60+ | 0.610 | 0.286 | 0.299 |
| Vocational Degree | Completed vocational internship/volunteer work | 0.640 | 0.491 | 0.278 |
| | Vocational school diploma | 0.600 | 0.314 | 0.264 |
| | University of applied sciences degree | 0.626 | 0.272 | 0.339 |
| | Specialist school diploma | 0.603 | 0.269 | 0.355 |
| | Completed apprenticeship | 0.618 | 0.304 | 0.305 |
| | Master craftsman or technician qualification | 0.626 | 0.297 | 0.374 |
| | University degree | 0.618 | 0.287 | 0.334 |
| | In vocational training | 0.648 | 0.238 | 0.306 |
| | Commercial or agricultural apprenticeship | 0.629 | 0.335 | 0.287 |
| | Commercial apprenticeship | 0.630 | 0.302 | 0.320 |
| | No vocational training completed | 0.619 | 0.289 | 0.242 |
| Gender | Male | 0.617 | 0.300 | 0.325 |
| | Female | 0.619 | 0.279 | 0.279 |
| Leaning Party | AfD | 0.618 | 0.329 | 0.351 |
| | CDU/CSU | 0.612 | 0.325 | 0.301 |
| | FDP | 0.648 | 0.374 | 0.423 |
| | Grünen | 0.639 | 0.232 | 0.556 |
| | A minor party | 0.606 | 0.307 | 0.323 |
| | Linke | 0.594 | 0.256 | 0.267 |
| | SPD | 0.619 | 0.275 | 0.318 |
| | No party | 0.651 | 0.340 | 0.352 |
| Region | East Germany | 0.599 | 0.300 | 0.294 |
| | West Germany | 0.623 | 0.285 | 0.299 |
| Education Degree | High school diploma | 0.617 | 0.279 | 0.326 |
| | Higher education entrance qualification | 0.622 | 0.293 | 0.343 |
| | Secondary school diploma | 0.625 | 0.320 | 0.241 |
| | Intermediate school diploma | 0.619 | 0.295 | 0.302 |
| | Student | 0.833 | 0.680 | 0.379 |
| | No school diploma | 0.667 | 0.385 | 0.348 |

Table 14: Detailed JS Distances in each social group category in Experiment 1

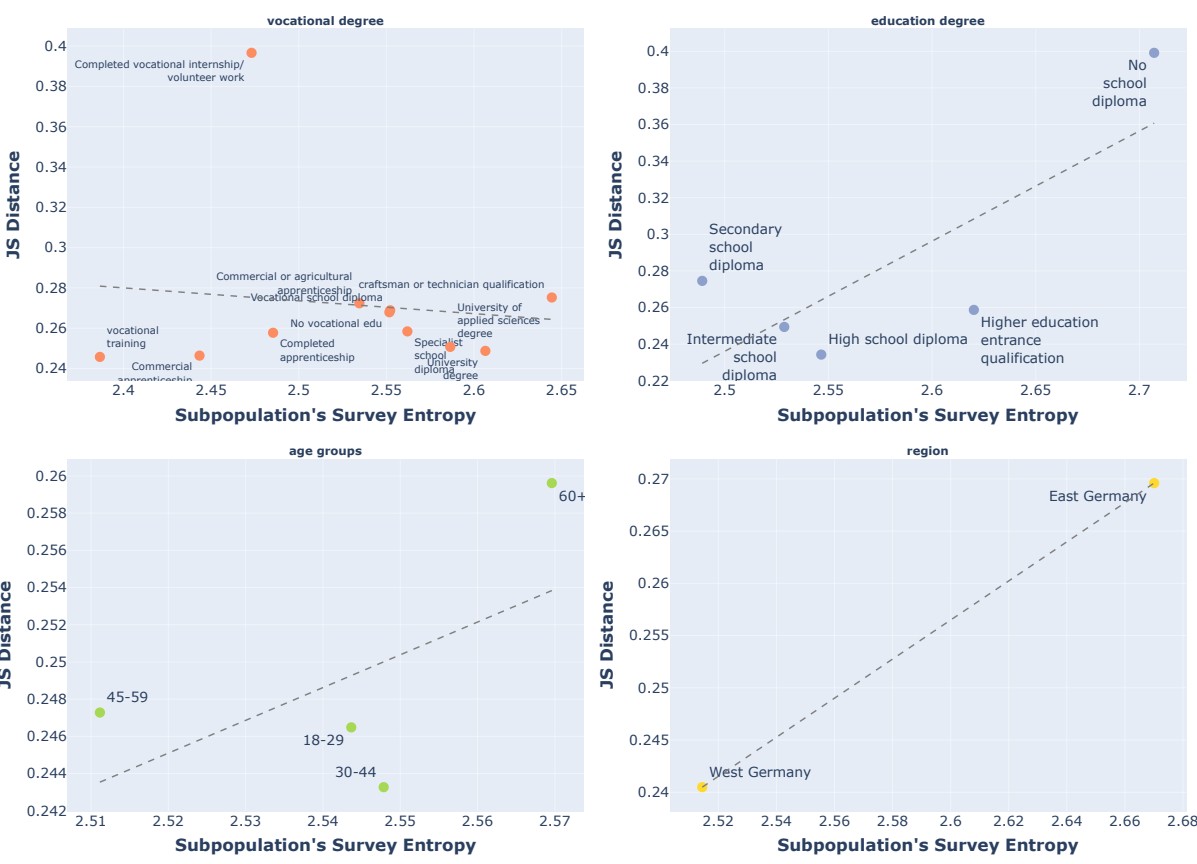

Figure 15: Survey's Subpopulation Entropy and the JS Distance in Experiment 2 (mean values for waves 12-21).

| Prompt | Llama2 Answer |
|---|---|
| What were the vote percentages for the major candidates in the 2016 US presidential election? | In the 2016 US presidential election, the vote percentages for the major candidates were:

Donald Trump (Republican): 46.1%
Hillary Clinton (Democratic): 48.2%
Gary Johnson (Libertarian): 3.3%
Jill Stein (Green): 1.1%

Note: These figures represent the national popular vote, excluding Maine and Nebraska, which allocate electoral votes proportionally. |
| What were the vote percentages for the major candidates in the 2020 US presidential election? | In the 2020 United States presidential election, the vote percentages for the major candidates were as follows:
* Joe Biden (Democratic Party): 51.1% (665,8533,295 votes)*
Donald Trump (Republican Party): 47.5% (62,2045,637 votes)*
Other candidates: 1.4% (19,362,8822 votes)
Note: These figures are based on the official results from the Federal Election Commission and may not reflect final totals or percentages. |
| What were the vote percentages in the 2017 German federal election? | The vote percentages in the 2017 German federal election were as follows:

Chancellor Angela Merkel's Christian Democratic Union (CDU): 32.9%
Social Democratic Party (SPD): 20.5%
Alternative for Germany (AfD): 12.6%
Free Democratic Party (FDP): 10.7%
Green Party: 9.2%

Note: These figures represent national vote share. Actual seat distribution may differ due to proportional representation. |
| What is the most important problem in Germany in November 2019? List the answers and their percentages for the German population. Do your best guess. | It is difficult to determine the most important problem in Germany in November 2019, as opinions and priorities can vary greatly among individuals and groups. However, based on current events and societal trends, here are some of the issues that might be considered important by the German population:
1. Climate change and environmental protection (25-30%) - Concerns about climate change and the environment have been growing in Germany, with increased activism and calls for stronger government action to reduce carbon emissions and protect natural resources.
2. Economy and job security (20-25%) - The German economy has been facing challenges, including a slowdown in growth and job insecurity, particularly in the automotive and manufacturing sectors.
3. Immigration and integration (15-15%) - Germany has been a destination for many refugees and migrants, and the integration of these individuals into society has been a persistent challenge, with debates around issues such as employment, education, and social cohesion.
4. Healthcare and social welfare (5-10%) - Germany's healthcare system has been under strain, with debates over funding and access to certain medical services, particularly for low-income families.
5. European and international politics ( < 5%) - Germany has played a key role in European Union |

Table 15: Checking the factual knowledge of Llama2 model without giving any survey contexts

