# OpenReview forum: "Algorithmic Fidelity of Large Language Models in Generating Synthetic German Public Opinions: A Case Study"
_colmweb.org/COLM/2025/Workshop/Social_Sim — Social Sim'25_

### Official Review · Reviewer_YRVV · 2025-07-16
**A study of LLM representation of German public opinion.**

**Rating:** 5
**Overall Assessment:** 3
**Confidence:** 3

**Review:**

This paper offers an interesting exploration of algorithmic fidelity in LLMs modeling German public opinion, with a well-presented experimental setup and some relevant analyses. However, I feel that the choice of models feels somewhat outdated and under-justified, and the lack of experimentation with newer and/or proprietary models and comparisons is a limitation. The results are promising but largely serve to confirm prior findings, and several important questions, like the effect of fine-tuning or model safeguards, remain unaddressed. As it stands, the paper reads mostly as a prompting evaluation with some statistical analysis, leaving room to enhance both originality and significance. I believe this work has potential, if the authors can build on reviewer feedback to round out the presentation.

**Comments Suggestions And Typos:**

The authors discuss the “framing effect” but do not introduce or explain it for readers who are not familiar with the effect, such as myself. A small introductory sentence prior to this discussion would be helpful.

Small optional aesthetic note, I think the following two sentences would flow much better and be easier to read if they were combined into one sentence, and if the sentence did not begin with ‘And’, but instead something else like ‘Further’, ‘Additionally’, etc:

“And no age variable value consistently has a 188 lower JS score. It shows the model can represent the opinions of different social groups at various 189 levels but offers no clear explanation for the variation in representation.”

Also adding some more v-space between Figure 8 and its caption.

Formula 6 ending where authors cut back to discussion is ungrammatical and not well formatted: Remove space before “here” (same with “However” at start of next paragraph), and replace “here” with “where”. Add a comma at the end of the formula itself.

“The peak of wave 21 compared to wave 20 corresponds to…”; did the authors mean “The peak of wave 20 compared to wave 21 corresponds to…”, since wave 20 sees the peak, not wave 21?

**Paper Summary:**

The authors examine the alignment between LLM representations of public opinion, particularly in Germany in this study, compared to actual data—this is called the algorithmic fidelity of an LLM. They study Llama-2, Gemma, and Mixtral over a set of three experiments evaluating how algorithmic fidelity is affected by temporal context, among other related variables. The first experiment analyzes, on a single moment in time, how well the LLMs represent dataset statistics, finding that Llama-2 shows less information gain, and aligns more closely with survey responses. The second experiment studies how, throughout many points in time, LLM performance improves on representing relative population data; the authors take the Pearson correlation between survey entropy and JS distance, and find that (a) JS distance decreases over survey waves, and (b) Llama-2 sees proportionally reduced population representativeness with increased response diversity. The third experiment probes which variables are most impactful on representation and response diversity, finding that the political party variable has the greatest effect on performance. The authors end with some interesting future recommendations:

1. LLMs should be capable of understanding and representing diverse, nuanced views and identities.
2. English-centric LLM evaluations are not sufficiently inclusive for all human identities.
3. As LLMs are updated more often than some public opinion data and records, there is potential for their usage within estimating shifts in opinion

**Relevance:**

4

**Summary Of Strengths:**

The paper presents a good literature review, with lots of interesting and relevant papers cited throughout. Additionally, while the introduction comes off as slightly awkward to read, the methods and experiments portions are much cleaner.

**Summary Of Weaknesses:**

It would strengthen the paper if the authors justify their choice in models, as I have compiled a list of questions which crossed my mind throughout reading:
Why were no proprietary models tested, as no fine-tuning was necessary for the experiment?
Why didn’t the authors examine fine-tuning the open models on some survey data?
Most of the LLMs used have had updated versions for many years now, such as Llama and Gemma; why have the authors not decided to probe newer model versions, such as Gemma-2 or 3, and Llama-3+?
Why did the authors not study how base and instruction-tuned models differ in algorithmic fidelity? This could have answered some interesting questions on the effect of human-preference alignment on fidelity.

Further, I have some notes and questions from each of the experiments:

Experiment 1: Table 1 is a bit hard to interpret—why is COVID included? Just to show that Gemma will be excluded from further experiments? Is there no more recent data that can be analyzed?

Experiment 2: Llama-2 models left-leaning groups better than right-leaning; how much of these results are affected by limitations/biases introduced by model safeguards, particularly with parties like AfD?

Experiment 3: I feel that the first two paragraphs were not interesting to me as findings; what was more interesting from the figure introduced in this portion was which variables were most impactful, which is addressed in the third paragraph. I believe authors focusing discussion in this portion more to this angle, and prioritizing this discussion first in the section could be a better use of space.

---

### Meta-Review · Area_Chair_yp6s · 2025-07-21

**Recommendation:** Accept

**Metareview:**

--